# Projection of dengue fever transmissibility under climate change in South and Southeast Asian countries

**Yawen Wang[1], Conglu Li[1], Shi Zhao[1,2,3,4], Yuchen Wei[1,2], Kehang Li[1], Xiaoting Jiang[1], Janice Ho[5], Jinjun Ran[6], Lefei Han[7], Benny Chung-ying Zee[1,2,3], Ka Chun Chong[1,2,3]***

**1** Jockey Club School of Public Health and Primary Care, The Chinese University of Hong Kong, Hong Kong Special Administrative Region, China, **2** Centre for Health Systems and Policy Research, The Chinese University of Hong Kong, Hong Kong Special Administrative Region, China, **3** Clinical Trials and Biostatistics Laboratory, Shenzhen Research Institute, The Chinese University of Hong Kong, Shenzhen, China, **4** School of Public Health, Tianjin Medical University, Tianjin, China, **5** Division of Landscape Architecture, Department of Architecture, Faculty of Architecture, The University of Hong Kong, Hong Kong, Hong Kong Special Administrative Region, China, **6** School of Public Health, Shanghai Jiao Tong University School of Medicine, Shanghai, China, **7** School of Global Health, Chinese Center for Tropical Diseases Research, Shanghai Jiao Tong University School of Medicine, Shanghai, China

* marc@cuhk.edu.hk

**Data Availability Statement:** All data used for this study are publicly available. The links to data sources are provided in S1 Table.

## Abstract

Vector-borne infectious disease such as dengue fever (DF) has spread rapidly due to more suitable living environments. Considering the limited studies investigating the disease spread under climate change in South and Southeast Asia, this study aimed to project the DF transmission potential in 30 locations across four South and Southeast Asian countries. In this study, weekly DF incidence data, daily mean temperature, and rainfall data in 30 locations in Singapore, Sri Lanka, Malaysia, and Thailand from 2012 to 2020 were collected. The effects of temperature and rainfall on the time-varying reproduction number ($R_t$) of DF transmission were examined using generalized additive models. Projections of location-specific $R_t$ from 2030s to 2090s were determined using projected temperature and rainfall under three Shared Socioeconomic Pathways (SSP126, SSP245, and SSP585), and the peak DF transmissibility and epidemic duration in the future were estimated. According to the results, the projected changes in the peak $R_t$ and epidemic duration varied across locations, and the most significant change was observed under middle-to-high greenhouse gas emission scenarios. Under SSP585, the country-specific peak $R_t$ was projected to decrease from 1.63 (95% confidence interval: 1.39–1.91), 2.60 (1.89–3.57), and 1.41 (1.22–1.64) in 2030s to 1.22 (0.98–1.51), 2.09 (1.26–3.47), and 1.37 (0.83–2.27) in 2090s in Singapore, Thailand, and Malaysia, respectively. Yet, the peak $R_t$ in Sri Lanka changed slightly from 2030s to 2090s under SSP585. The epidemic duration in Singapore and Malaysia was projected to decline under SSP585. In conclusion, the change of peak DF transmission potential and disease outbreak duration would vary across locations, particularly under middle-to-high greenhouse gas emission scenarios. Interventions should be considered to slow down global warming as well as the potential increase in DF transmissibility in some locations of South and Southeast Asia.

**Funding:** This work was partially supported by Health and Medical Research Fund (Author: KCC; grant numbers INF-CUHK-1; https://rfs1. healthbureau.gov.hk/english/funds/funds_hmrf/ funds_hmrf_abt/funds_hmrf_abt.html). The funding agencies had no role in the design and conduct of the study; collection, management, analysis, and interpretation of the data; preparation, review, or approval of the manuscript; or decision to submit the manuscript for publication.

**Competing interests:** The authors have declared that no competing interests exist.

## Author summary

Climate change has a significant impact worldwide, including the proliferation of infectious diseases in tropical areas. While the warmer and more humid living conditions in these regions are generally considered favorable for the transmission of vector-borne diseases, we argue that the effects of climate change on DF infection may vary across different locations. Unlike previous studies focusing primarily on the disease incidence, this study aims to estimate changes in peak DF transmissibility and the duration of disease epidemics in the future. By analyzing data from 30 locations in four tropical countries, we have found that changes in DF transmissibility and epidemic duration exhibit variations both within and between countries, particularly under the scenarios of moderate to high greenhouse gas emissions. In light of these findings, we recommend conducting location-specific evaluations of DF transmissibility under the influence of climate change, as well as implementing targeted interventions to mitigate the potential increase in DF transmission rates in specific tropical locations.

## Introduction

The accelerated global warming resulting from human activity-induced greenhouse gas emissions has been a prominent issue since last century. This continuous upward trend is believed to be responsible for an unprecedented increase in global surface temperature and more frequent extreme weather events such as heatwaves in the future [1,2]. Climate change with global warming is now regarded as a severe public health threat due to the increased morbidity and mortality risks of diseases such cardiovascular disease and mental health problems with elevated temperatures [3–5]. Furthermore, climate change is also considered as a risk factor that affects the transmission dynamics, geographic spread, and resurgence of infectious diseases, particularly those transmitted by vectors [6].

Dengue fever (DF), a vector-borne disease that is sensitive to climate, shows year-round epidemics in most tropical countries [7,8]. Among the DF hotspots, South and Southeast Asian countries account for one-third of the global DF burden, and the DF incidence in these regions has increased by over 46% in the last five years [9]. This remarkable increase may be partially attributed to the warm climate that provides a suitable living environment for DF vectors. Research conducted in laboratories and fields has demonstrated the crucial role of ambient temperature in regulating the behaviors of these vectors. A rise in temperature is associated with an increased vector-host virus transmissibility, as well as a shortened extrinsic incubation period [10,11]. These temperature-dependent traits may be responsible for the increased risk of DF infection and its expansion over recent decades. In addition, the evenly distributed rainfall in this region may create ample breeding sites for DF vectors, resulting in a larger vector population and an increased DF spread risk.

Climate change is expected to have an impact on future DF epidemics, with rising global temperatures and rainfall playing a significant role [12,13]. Greenhouse gas emission is generally admitted as one of the key contributors to climate change, and the future climates are usually divided into low, middle, or high emission groups, which represent slow, middle, and rapid changes of temperature, rainfall, and other climate variables, respectively [14]. Most studies anticipate a rapid increase in DF transmissibility under high emission scenario, leading to more frequent outbreaks by the end of this century [15,16]. Conversely, only a slight increase in the DF transmissibility is projected under the low carbon emission scenario [17].

However, several studies suggest that rising temperatures may cause a pronounced decline in DF epidemics once the temperature exceeds the vectors' threshold [18]. Additionally, studies indicate a geographical expansion of DF in the future, potentially driven by the increased temperature and rainfall, which would shift the currently non-epidemic areas to suitable habitats for the DF vector [13]. Despite the inconsistent projections regarding DF transmission, previous research on climate change and its impact on DF transmission primarily focused on subtropical or temperate areas, with less evaluation conducted in tropical areas [19].

Considering the limited studies investigating the disease spread under climate change in South and Southeast Asia [20,21], this study aimed to project the DF transmission potential in 30 locations across four South and Southeast Asian countries under different climate change scenarios. The findings are expected to improve the understanding of the DF spread potential under climate change, and thus to support DF prevention and control in the future.

## Methods

### Ethics statement

Ethics approval is not required as only retrospective aggregated data were used.

### Study setting and case data

South and Southeast Asia countries were screened for publicly available city-level or province-level officially released daily or weekly DF case data. A total of 30 locations in four countries (i.e., Singapore, Sri Lanka, Malaysia, and Thailand) were included, and weekly DF data from 2012 to 2020 was collected from official websites [22]. Data sources are listed in S1 Table.

### Meteorological data

Location-specific daily mean temperature (˚C) and rainfall (mm) from 2012 to 2020 were collected from the National Centers for Environmental Information (NCEI), which provides station-based records. At least one weather station in a location was included in this study to retrieve daily meteorological data. The mean value over all weather stations was used to represent the overall weather condition in cases where a location had multiple available weather stations. Location-specific daily mean temperatures in each week were averaged to obtain the weekly mean temperature, and the daily rainfall was summed up to calculate the weekly total rainfall.

### Projected data under Shared Socioeconomic Pathways

Projected climate data was collected from WorldClim 2.0 (http://www.wordclim.org), which provides the downscaled and bias-corrected gridded monthly averages of maximum and minimum temperatures, and monthly cumulative rainfall. Climate projections were collected from 11 Global Climate Models (GCMs) developed in the Coupled Model Intercomparison Project 6 (CMIP6) report (see S2 Table for details), and the model selection was inspired by previous research conducted in South and Southeast Asia [23–25]. Data in four periods, namely 2030s (2021 to 2040 average), 2050s (2041 to 2060 average), 2070s (2061 to 2080 average), and 2090s (2081 to 2100 average) under three Shared Socioeconomic Pathways (SSPs) were collected (SSP126, SSP245, and SSP585) with a resolution of 2.5 minutes (~21 km$^2$ at the equator). The SSP reflects the future economic growth and demographics that could affect global greenhouse gas emissions and mitigations. SSP126 represents a relatively low pathway for future greenhouse gas emission scenarios with an additional radiative forcing of 2.6 W/m$^2$ by the year 2100, while SSP245 represents a medium pathway with a value of 4.5 W/m$^2$ by the year 2100. Similarly, SSP585 represents the upper boundary of the emission scenario with 8.5 W/m$^2$ by the end

of this century [14]. The projected monthly maximum temperature and minimum temperature were averaged to get the projected monthly mean temperature. Temperature and rainfall projections from all 11 models were averaged to get the overall monthly projected values in each period. Methods used to project weekly temperature and rainfall were detailed in S1 Text. Projected population data under different SSP scenarios was collected from Wang et. al [26].

## Time-varying reproduction number estimation

The time-varying reproduction number ($R_t$) serves as the metric of disease transmissibility, representing the average number of secondary infections generated by an infected person at a time point during a disease outbreak. If $R_t$ is greater than one, the disease is spreading and the outbreak is likely to sustain. Typically, $R_t$ is derived from the time series of infected cases and the serial interval [27]. In this study, the $R_t$ was estimated by the renewal equation:

$$R_t = \frac{I_t}{\sum_{k=1}^{t} w_k I_{t-k}}$$

where $I_t$ denotes the new DF cases at week $t$, the $w_k$ is the probability distribution of serial interval equal to $k$ number of days, and the denominator represents the total infectiousness at time $t$. We assumed a serial interval distribution with mode equal to 18 days under 28˚C based on Siraj et al.[28], and discretized it into a weekly scale. In addition, the DF epidemic duration was defined as the number of weeks with weekly $R_t$ higher than unity in a year.

## Projection of location-specific $R_t$

Generalized additive models (GAM) were fitted to predict the location-specific $R_t$. In each location, a variety of GAMs with different numbers of lags of temperature and rainfall were included. Short-term trends and autocorrelation were also accounted for in the model. The GAM model can be expressed as follows:

$$E(g(R_t)) = \alpha + \sum_{i=0}^{n} f(Temp_{t,lagi}) + \sum_{j=0}^{n} f(Rain_{t,lagj}) + f(week_t) + \delta_t$$

where $g(.)$ is the log link function, $E(g(R_t))$ is the expectation of the $R_t$ in week $t$, $\alpha$ is the intercept. $Temp_{t,lagi}$ and $Rain_{t,lagj}$ represent the weekly mean temperature and total rainfall with $i$ and $j$ weeks lag at week $t$, with a maximum lag of $n$ weeks ($n$ = 0, 1, 2, 3, 4, 5, 6). $f()$ is the spline function for climate variables. $week_t$ is the week number in a year, serving as a proxy of seasonality. $\delta_t$ represents the autocorrelation. Models accounted for population were also established, and the results are reported separately.

Weekly dengue fever data from 2012 to 2017 was used to train the model. Data from 2018 to 2020 was then used to validate the prediction performance of the model. A total of 49 models were constructed in each location, and the model with the lowest Root Mean Square Error (RMSE) and highest correlation coefficients in model fitting and forecasting was selected to project the location-specific $R_t$ from 2030s to 2090s under different SSP scenarios. By using the projected series of input variables, the weekly $R_t$ series in the future will be estimated for each location under each climate change scenario and time-period. Two indicators, peak $R_t$ and epidemic duration, were assessed to quantitatively measure the disease transmission potential. Epidemic duration was determined as number of weeks with weekly $R_t$ higher than one in a given year. All calculations and plots were completed by R 4.0.2 with *raster* and *mgcv* packages. Maps were created using the *rnaturalearth* and *rnaturalearthdata* packages, which utilize map data from the publicly available dataset Natural Earth (https://www.naturalearthdata.com/).

## Results

The 30 study locations reported a total of 1,231,874 cases from 2012 to 2020 (Fig 1 and S3 Table and S4 Table). The country-specific nine-year averaged DF incidence was around 25.00 cases per 10,000 people in Singapore, Sri Lanka, and Malaysia, while Thailand had a significantly lower DF incidence rate of 18.21 cases per 10,000 people. Across the 30 included locations, the highest disease incidence (72.35 per 10,000 people) was estimated at Selangor, Malaysia, while the lowest rate (6.56 per 10,000 people) was found at Sarawak, Malaysia. The nine-year average weekly rainfall ranged from 26.86mm (interquartile range, IQR = 38.93mm) in Thailand to 47.47mm (IQR = 54.07mm) in Malaysia. The highest and lowest weekly total rainfall were 69.00mm in Ratnapura and 19.90mm in Hambantota, Sri Lanka. The mean temperature was around 27.00˚C in these locations, except Badulla that had an average mean temperature of 23.10˚C. The mean temperature was projected to be around 30.00˚C in 2090s under SSP585, accompanied by an increase of approximately 10mm in weekly rainfall for these countries.

During 2012–2020, settings in Sri Lanka and Thailand exhibited more pronounced $R_t$ fluctuations compared to that in Singapore and Malaysia (Figs 2 and S1). The estimated location-averaged $R_t$ in Thailand usually peaked at the middle of each year with the peak value exceeding 2.00. Sri Lanka had two $R_t$ peaks per year, with peak values lower than those in Thailand. The highest $R_t$ in Thailand and Sri Lanka was 3.31 and 3.03 in week 22 of 2014, surpassing those of the other two countries. $R_t$ in Singapore reached a peak of 2.15 in week 18 of 2019, while the highest value in Malaysia was observed as 1.79 in week 26 of 2014.

Disease transmissibility in each location was projected using the best-fitted location-specific GAM model. Among the 30 models, the RMSEs for fitting and forecasting varied between 0.14 (Selangor, model with temperature up to 6 weeks lag and rainfall up to 6 weeks lag) and 1.01 (Galle, model with temperature up to 4 weeks lag and rainfall up to 6 weeks lag), while the correlation coefficients ranged from 0.10 (Hambantota, model with temperature up to 5 weeks lag and rainfall up to 4 weeks lag) to 0.86 (Chiang Rai, model with temperature up to 5 weeks lag and rainfall up to 6 weeks lag) (S3 Table).

The changes in peak $R_t$ in a year would vary across locations and countries in the future (Fig 3). Under SSP126, the changes in projected peak $R_t$ were slight, with the most significant increase from 2.50 (95% confidence interval [CI]: 1.59–3.94) in 2030s to 2.77 (1.55–4.93) in 2090s in Puttalam, Sri Lanka, and the most remarkable decline from 2.55 (1.87–3.46) in 2030s to 2.29 (1.65–3.17) in 2090s in Trincomalee, Sri Lanka. The changes of peak $R_t$ were more notable under SSP245 and SSP585. Under SSP585, 19 out of 30 locations would experience a gradual decline in peak $R_t$ over the course of this century, with the most remarkable decrease from 3.65 (2.98–4.60) in 2030s to 1.83 (0.84–3.99) in 2090s in Chiang Rai, Thailand. The remaining 11 locations would exhibit a steady increase in peak $R_t$, with the most significant increase from 2.58 (1.61–4.14) in 2030s to 4.38 (1.53–12.48) in 2090s in Puttalam, Sri Lanka. Similar changing trends were predicted once the population change was accounted, while the magnitude of the change would be higher, especially in Thailand and under high emission scenarios (S2 Fig)

The country-specific peak DF transmission potential was projected to have the most significant change under SSP585, with the peak $R_t$ decrease from 1.63 (95% CI: 1.39–1.91), 2.60 (1.89–3.57), and 1.41 (1.22–1.64) in 2030s to 1.22 (0.98–1.51), 2.09 (1.26–3.47), and 1.37 (0.83–2.27) in 2090s in Singapore, Thailand, and Malaysia, respectively. The peak $R_t$ in Sri Lanka changed slightly from 1.95 (1.42–2.69) in 2030s to 1.96 (1.02–3.77) in 2090s under SSP585 (Fig 3).

The change in epidemic duration was projected to vary across locations (Fig 4). Under SSP126, the epidemic duration in P.Pinang, Malaysia in 2090s declined by 9 weeks compared

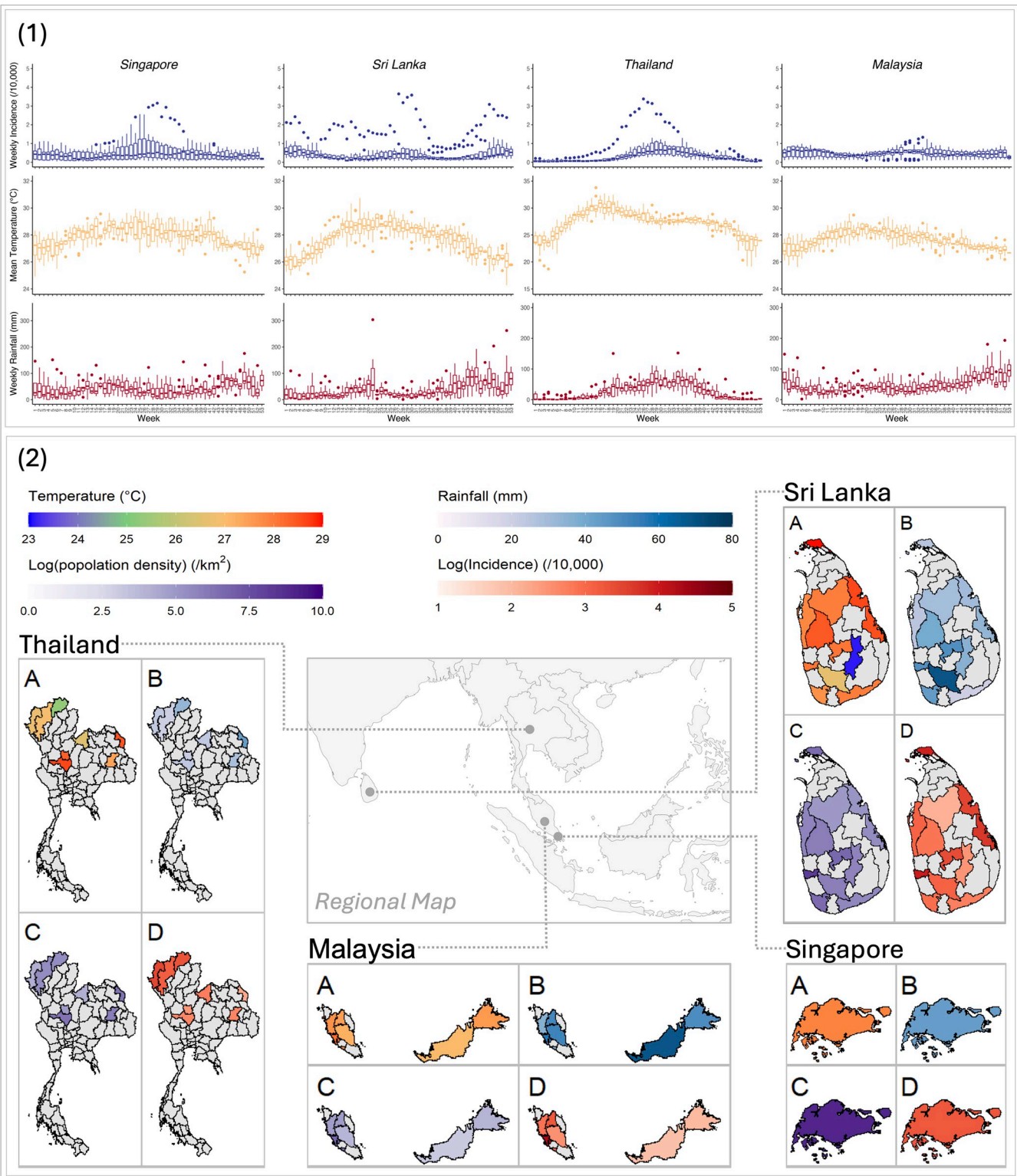

**Fig 1. Distribution of DF incidence, mean temperature, and weekly rainfall series in four countries.** Panel 1, weekly series box plots. Each box in the plot represents the location-averaged weekly values for the same week of each year across all included locations in the country. The line within the box denotes the median value, while the whiskers on the top and bottom of each box represent the maximum and minimum location-averaged values observed for that week over the span of nine years. Panel 2, spatial distribution of weekly mean temperature (A), weekly total rainfall (B), population density (C), and dengue fever incidence rate (D) in four countries. The map was obtained from Natural Earth, a public domain dataset (Term of Use: https://www.naturalearthdata.com/about/terms-of-use/) through the R packages *rnaturalearth* and *rnaturalearthdata*.

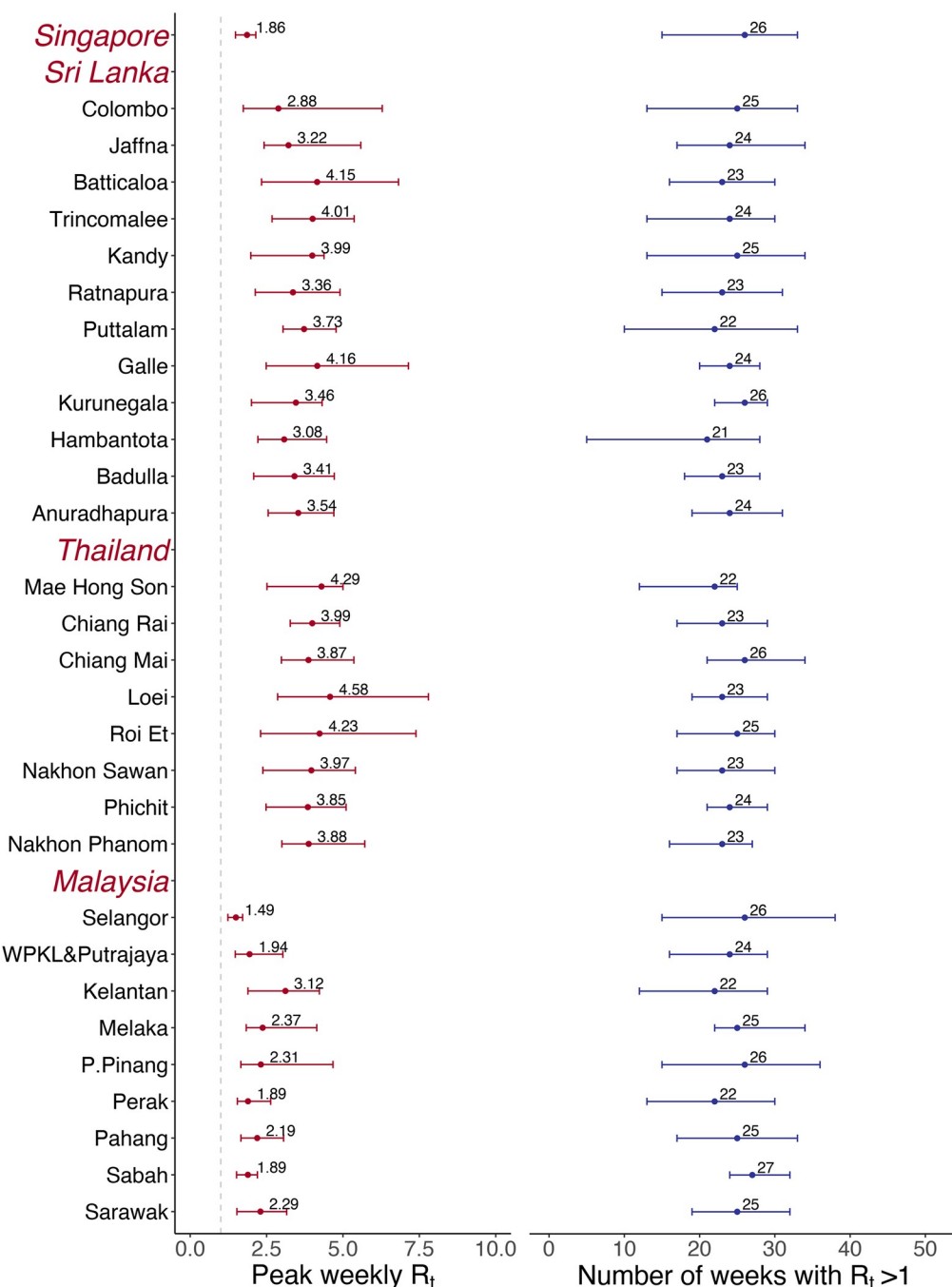

**Fig 2. Estimated mean and range of annual peak $R_t$ and epidemic duration.** DF epidemic duration was defined as number of weeks with $R_t$ higher than unity in a year. WPKL&Putrajaya: Wilayah Persekutuan Kuala Lumpur & Putrajaya. P.Pinang: Pulau Pinang. Locations within each country are ordered from the highest incidence to the lowest. Observed nine-year peak $R_t$ values are used to calculate the mean and range of annual peak $R_t$, while the epidemic durations of each of the nine years are used to calculate the mean and range of annual epidemic duration.

to 2030s, resulting in the most significant change among the 30 locations. Under SSP245, the epidemic duration in all of the nine locations in Malaysia would have an obvious decline. Particularly, P.Pinang and Selangor would experience a reduction of over 30 weeks in the 2090s compared to the 2030s. The most notable alteration in DF epidemic duration would be

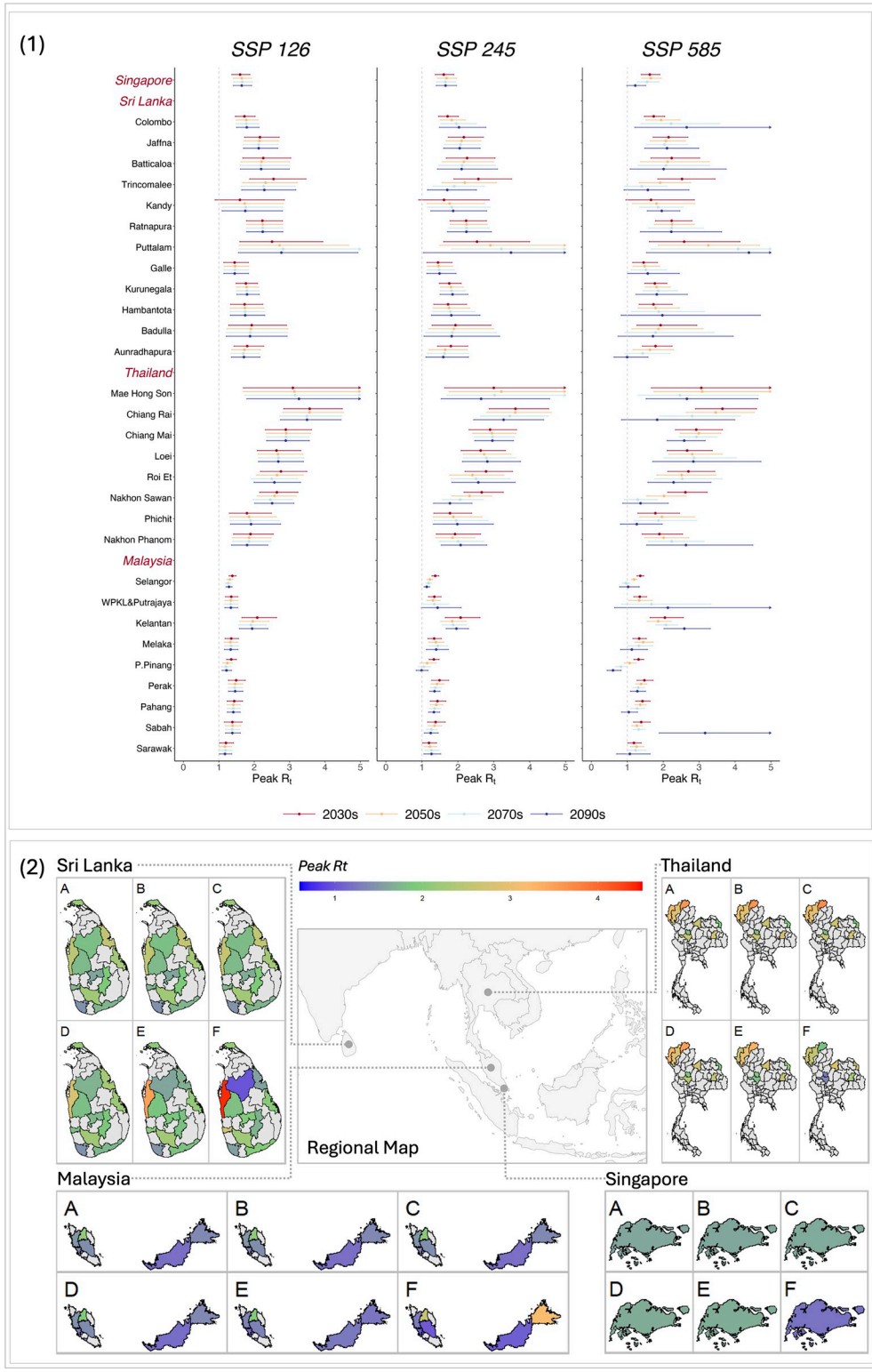

**Fig 3. Projection of the DF peak weekly $R_t$ in 30 locations in four countries.** Panel 1, projected peak weekly $R_t$ and 95% confidence interval of the included locations. Panel 2, special distribution of the projected peak weekly $R_t$ in the future: SSP126-2030s (A), SSP245-2030s (B), SSP585-2030s (C), SSP126-2090s (D), SSP245-2090s (E), and SSP585-2090s (F). WPKL&Putrajaya: Wilayah Persekutuan Kuala Lumpur & Putrajaya. P.Pinang: Pulau Pinang. Estimated by models without accounting for the population. The map was obtained from Natural Earth, a public domain dataset

(Term of Use: https://www.naturalearthdata.com/about/terms-of-use/) through the R packages *rnaturalearth* and *rnaturalearthdata*.

exhibited under SSP585. Compared to 2030s, the epidemic duration in Singapore and Malaysia kept declining in 2090s. However, the situation was inconsistent in Sri Lanka and Thailand regarding the changes in epidemic duration. For example, Kandy in Sri Lanka had an increase of 31 weeks in the epidemic duration between 2030s and 2090s. Including population results a similar trend but more remarkable change in DF epidemic duration (S3 Fig)

## Discussion

Projecting the infectious disease transmission potential under climate change is of great importance in light of the ongoing global warming. This study projected future temperature and rainfall-dependent DF transmission potential under three SSP scenarios with 11 GCMs for four periods centered at the 2030s, 2050s, 2070s, and 2090s. Primarily, our findings indicated that the change of DF peak transmissibility and disease epidemic duration in the future would vary across locations, with the most remarkable impact under a high emission scenario.

The peak DF transmission potential was projected to increase by up to 1.80 or decline by up to 1.82 under a high emission scenario, and the change would show within-country and between-country variations. Consistent with prior studies highlighting a decreasing trend in DF transmissibility, we projected a gradual decline in peak DF transmission potential in Singapore and Malaysia, which may be attributed to fewer favourable environmental conditions in these areas for DF spread in the future [29,30]. Prior studies suggested threshold effects of temperature and rainfall, meaning that once the temperature and rainfall exceed the vector's threshold, the less favourable living environment for mosquito survival would affect DF transmission [10]. Singapore and Malaysia have equatorial climates with high average temperatures and uniformly distributed rainfall throughout a year, resulting in a breeding ground for Aedes mosquitoes. However, the living environment may be a catastrophe to DF vectors once the temperature goes beyond the limits, ultimately leading to a declined DF transmissibility in these areas. Nevertheless, inconsistent results of an increased peak transmission risk were found in some locations in Sri Lanka and Thailand, especially in those with tropical wet and dry climates. The climate in these locations is characterized by distinct wet and dry seasons, and the projected increase in rainfall accompanied by the warmer temperature would create a more favorable environment for vector breeding and survival [31–33]. Additionally, changes of other non-climate factors such as demographic and economic situations may also affect the dengue fever transmission projections in the future [34,35].

We projected that the DF epidemic duration would decline under climate change in most locations, especially in Singapore and Malaysia. Consistent with the decreased peak DF transmissibility in certain locations, this study foresaw a shorter DF epidemic duration in the future. In contrast, it is worth noting that some locations in Sri Lanka and Thailand would have a longer epidemic duration in the future, with a more pronounced rising trend under SSP585. A global study has also reported a spatial heterogeneity in the duration of the DF epidemic, projecting both prolonged and shortened dengue fever transmission seasons across South and Southeast Asia by the end of this century [21]. It should be acknowledged that despite the predominant tropical weather in South and Southeast Asia, the diverse temperature and rainfall patterns within each region could lead to various climate patterns, consequently contributing to the inconsistent changing trend in DF epidemic period across locations.

Previous research emphasized a geographical expansion of DF in the future, indicating that areas with less noticeable DF epidemics may suffer from a more severe DF infection risk under

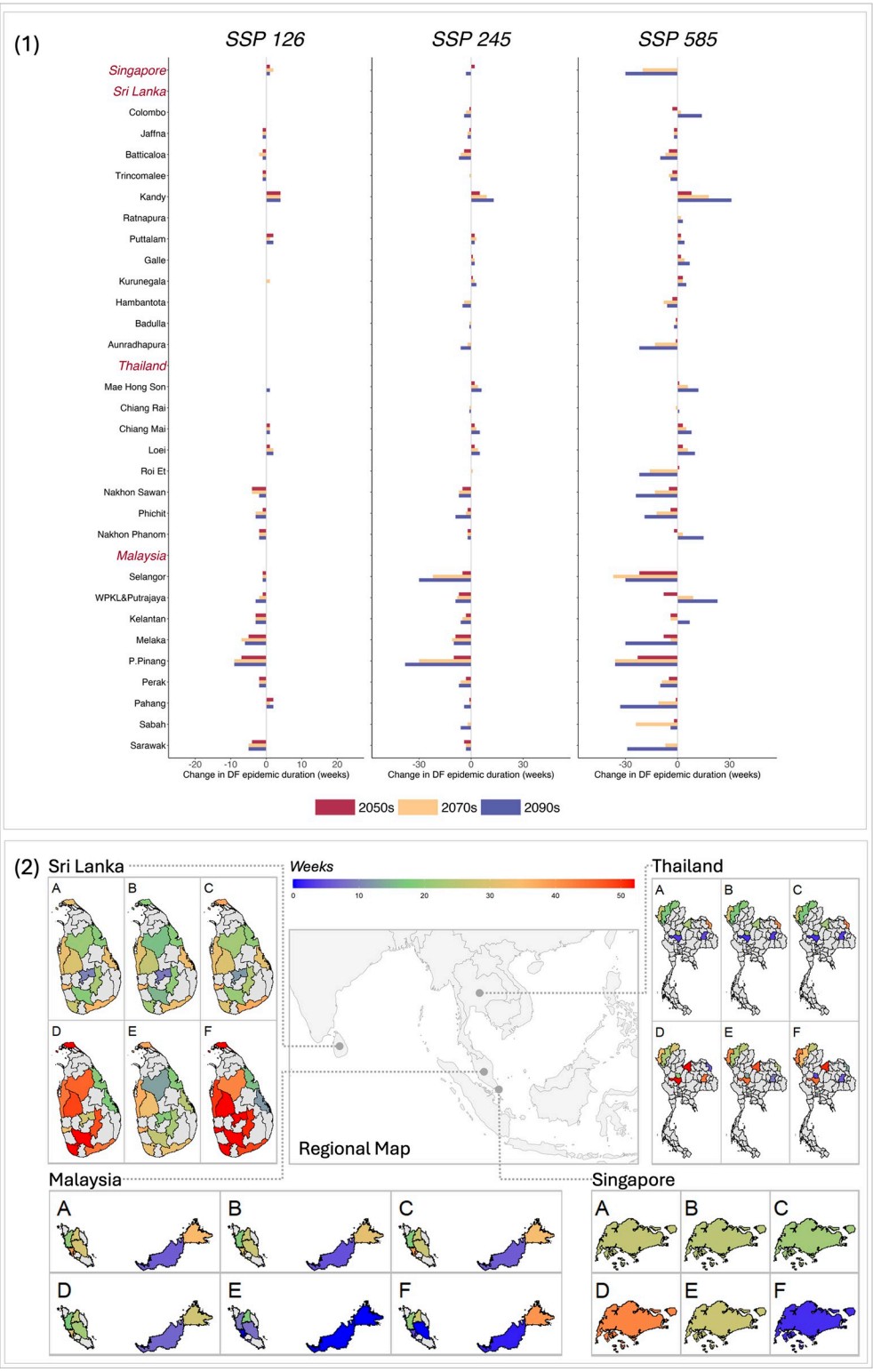

**Fig 4. Projected change of DF epidemic duration under climate change.** Panel 1, the projected change of DF epidemic duration in the future compared to the value in 2030s under each SSP scenario (S5 Table). DF epidemic duration was defined as number of weeks with $R_t$ higher than unity in a year. Panel 2, the special distribution of the projected DF epidemic duration in the future: SSP126-2030s (A), SSP245-2030s (B), SSP585-2030s (C), SSP126-2090s (D), SSP245-2090s (E), and SSP585-2090s (F). WPKL&Putrajaya: Wilayah Persekutuan Kuala Lumpur & Putrajaya. P.

Pinang: Pulau Pinang. Estimated by models without accounting for the population. The map was obtained from Natural Earth, a public domain dataset (Term of Use: https://www.naturalearthdata.com/about/terms-of-use/) through the R packages *rnaturalearth* and *rnaturalearthdata*.

climate change. This change pattern could partially be attributed to factors including cross-immunity or annual/interannual serotype fluctuation that increase the severity of the disease while the DF transmission declines [36]. Additionally, compared with tropical area with suitable living environment for DF vector, regions with relatively lower temperature and less rainfall are less likely to experience large DF epidemic. Nevertheless, more non-epidemic regions tend to become conducive for DF vector survival as global warming continues, leading to a high DF transmission risk and geographic extension in these areas in the future [17,37]. However, consistent with other research conducted in tropical regions, this research failed to show a noticeable increase in DF peak transmission potential or prolonged epidemic duration even in Thailand, which had the lowest temperature, rainfall, and DF incidence rate among the four countries [31]. The sustained high level of environmental suitability throughout this century in these tropical areas might contribute to the less apparent increase of DF risk in these regions.

Implications and interventions for DF prevention and control in South and Southeast Asia should also be mentioned. Firstly, the spatial heterogeneity of future changes in DF transmission potential highlights the necessity of estimating location-specific dengue infection risks and developing and imposing well-tailored disease prevention and control strategies. These may involve allocating additional healthcare resources to areas with higher DF transmission risk, as well as increasing investment in insecticide spraying and removing stagnant water. Secondly, projections of varying DF spread risks under different climate change scenarios highlight the sensitivity of DF transmission to climate fluctuations. Therefore, it is advisable to strengthen surveillance and monitoring efforts, as well as conduct dynamic assessments of DF infection risk to track the evolving nature of dengue dynamics and facilitate prompt response measures. Thirdly, the predicted warmer and wetter environments in specific regions may favor future DF transmission, emphasizing the significance of limiting global warming to a manageable range. The need for national and international regulations, such as reducing greenhouse gas emissions and promoting the use of clean and sustainable resources to mitigate climate change, is apparent. Lastly, collaborating with international partners to exchange best practices and coordinate efforts in addressing the global challenge of DF would also bring substantial benefits.

To our knowledge, there is limited research on estimating the impact of climate change on dengue fever transmissibility, specifically the reproduction number [15,38]. This study utilized location specific climate-dengue relationships to predict the disease transmission potential, revealing heterogeneous changes across locations. Previously, Wang et al. [39] has simulated the change of dengue fever incidence in four South and Southeast Asian locations by using compartmental models with temperature- and rainfall-dependent parameters, and suggested a potential increase in dengue fever peak incidence and longer disease epidemic duration in the future. While disease incidence data is useful for assessing the disease burden and tracking its progression, investigating the reproduction number as the transmissibility outcome complements this information by providing valuable insights into the transmission dynamics and potential of outbreaks.

Some limitations of this study should be acknowledged. Firstly, only temperature and rainfall were used as indicators of climate change to project DF transmission in the future, leaving other climate variables, such as wind speed, unaccounted for. Despite data availability, the

main reasons for this limitation are the well-documented temperature-/ rainfall-DF relationships, and the determined impact of these two variables on shaping DF transmission. Additionally, only population growth was considered in projecting future DF transmission potential due to the lack of other socioeconomic variables. We acknowledge that overlooking other influencing factors may reduce the accuracy of our estimation [40]. While the results of this study are promising as we focused primarily on the changes in dengue fever transmission potential throughout this century, more comprehensive estimations are needed to enhance our understanding of future DF infection risks. Furthermore, while this study employed standardized data sources and methods to mitigate the effects of differing surveillance efforts, potential variations in reporting bias among locations were not addressed due to a lack of available data, and this may reduce the precision of the findings. Another limitation arises from the selection of study locations, which were primarily determined by data availability. The diverse socioeconomic characteristics and other factors present in both the included and excluded locations may limit the overall representativeness of each country's landscape. Nonetheless, with using multi-settings, the study's findings can offer valuable insights into the changing trends of dengue transmissibility under the influence of climate change. In addition, the GCMs selection was inspired by previous studies, and we acknowledge the potential uncertainties in these models for predicting future climate. This study used the most recent climate projections based on CMIP6, whereas an updated analysis may be needed in the future to maintain the accuracy of the prediction. Moreover, this study used the 20-years averaged climate projections (e.g., 2030s referred to 2021–2040 average) to depict the general climate change condition over that period. This approach may overlook variations in climate and other fluctuations throughout the timeframe. It is recommended that future research could consider incorporating fine-scale yearly climate projections for more precise and detailed estimations of dengue fever transmission.

## Conclusion

The change of DF transmission potential in South and Southeast Asia was projected to vary across locations, and the most significant change was expected under a middle-to-high greenhouse gas emission scenario. Limiting global warming within a reasonable range would be meaningful for DF control. In addition, the variations in DF transmissibility projections suggests that the location-specific DF warning system needs to be implemented to enable more targeted and timely disease prevention and control interventions.

## Supporting information

**S1 Text. Supplementary methodology.**
(DOCX)

**S1 Table. Data sources.**
(DOCX)

**S2 Table. List of 11 included GCMs.**
(DOCX)

**S3 Table. Basic characteristics and model performance in 30 study locations.**
(DOCX)

**S4 Table. DF epidemiology, temperature, and rainfall conditions in four countries.**
(DOCX)

**S5 Table. Projected DF epidemic duration in 2030s under climate change.**
(DOCX)

**S1 Fig. Weekly DF $R_t$ in four countries from 2012 to 2020.**
(TIFF)

**S2 Fig. Estimation of the peak weekly $R_t$ and 95% confidence interval of DF in 30 locations in four countries.** WPKL&Putrajaya: Wilayah Persekutuan Kuala Lumpur & Putrajaya. P. Pinang: Pulau Pinang. Estimated by models accounting for the population.
(TIFF)

**S3 Fig. Estimated change of DF epidemic duration under climate change.** DF epidemic duration was defined as number of weeks with $R_t$ higher than unity in a year. WPKL&Putrajaya: Wilayah Persekutuan Kuala Lumpur & Putrajaya. P.Pinang: Pulau Pinang. Estimated by models accounting for the population.
(TIFF)

## Author Contributions

**Conceptualization:** Ka Chun Chong.

**Formal analysis:** Yawen Wang, Shi Zhao, Yuchen Wei.

**Funding acquisition:** Ka Chun Chong.

**Investigation:** Shi Zhao, Yuchen Wei.

**Methodology:** Yawen Wang, Shi Zhao, Yuchen Wei, Benny Chung-ying Zee, Ka Chun Chong.

**Software:** Shi Zhao, Yuchen Wei.

**Supervision:** Ka Chun Chong.

**Visualization:** Yawen Wang.

**Writing – original draft:** Yawen Wang, Conglu Li, Kehang Li, Ka Chun Chong.

**Writing – review & editing:** Shi Zhao, Yuchen Wei, Xiaoting Jiang, Janice Ho, Jinjun Ran, Lefei Han, Benny Chung-ying Zee.

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
