## [Decision Letter · Decision Letter 0]

6 Feb 2024

Dear Dr. Chong,

Thank you very much for submitting your manuscript "Projection of dengue fever transmissibility under climate change in South and Southeast Asian countries" for consideration at PLOS Neglected Tropical Diseases. As with all papers reviewed by the journal, your manuscript was reviewed by members of the editorial board and by several independent reviewers. In light of the reviews (below this email), we would like to invite the resubmission of a significantly-revised version that takes into account the reviewers' comments. 

Thank you for your patience with the review of this manuscript. The reviewers have made a number of constructive suggestions both on your methods and the explanation of your methods and result, which we encourage you to carefully consider in your revision. 

Additionally, we ask you to clarify, both in the manuscript and your response to reviewer comments, how the submitted mansucript is distinct from Wang et al 2023 Impact of climate change on dengue fever epidemics in South and Southeast Asian settings: a modeling study. 

Please note that comments from reviewer 2 are provided as an attachment

We cannot make any decision about publication until we have seen the revised manuscript and your response to the reviewers' comments. Your revised manuscript is also likely to be sent to reviewers for further evaluation.

Sincerely,

Elizabeth J Carlton

Academic Editor

Michael Holbrook

Section Editor

Thank you for your patience with the review of this manuscript. The reviewers have made a number of constructive suggestions both on your methods and the explanation of your methods and result, which we encourage you to carefully consider in your revision. 

Additionally, we ask you to clarify, both in the manuscript and your response to reviewer comments, how the submitted mansucript is distinct from Wang et al 2023 Impact of climate change on dengue fever epidemics in South and Southeast Asian settings: a modeling study. Please note that comments from reviewer 2 are provided as an attachment

Reviewer's Responses to Questions

**Key Review Criteria Required for Acceptance?**

**Methods**

-Are the objectives of the study clearly articulated with a clear testable hypothesis stated?

-Is the study design appropriate to address the stated objectives?

-Is the population clearly described and appropriate for the hypothesis being tested?

-Is the sample size sufficient to ensure adequate power to address the hypothesis being tested?

-Were correct statistical analysis used to support conclusions?

-Are there concerns about ethical or regulatory requirements being met?

Reviewer #1: Surveillance efforts will vary greatly across the four countries and local sites, and it does not appear the authors have taken this into account in their analyses. Although the sources are listed in the SI, there is no mention of potential reporting bias anywhere in the manuscript. Making cross-country comparisons is therefore problematic.

The authors state as a limitation that they only consider temperature/rainfall data in their models. Why is this the case? Although socio-economic variables are difficult to obtain projections for at a high spatial resolution, at the very least factors such as population density will have an effect on the varying incidence rates by site and need to be accounted for.

Reviewer #2: (No Response)

**Results**

-Does the analysis presented match the analysis plan?

-Are the results clearly and completely presented?

-Are the figures (Tables, Images) of sufficient quality for clarity?

Reviewer #1: The graphs are useful for showing error/uncertainty by country/site, but the paper would greatly benefit from having data displayed in map form (including input variables, incidence rates, and model outputs).

Reviewer #2: (No Response)

**Conclusions**

-Are the conclusions supported by the data presented?

-Are the limitations of analysis clearly described?

-Do the authors discuss how these data can be helpful to advance our understanding of the topic under study?

-Is public health relevance addressed?

Reviewer #1: The limitations are very lightly touched upon. What other variables should be considered besides temperature/rainfall and why have they not been in the current analysis?

Reviewer #2: (No Response)

**Editorial and Data Presentation Modifications?**

Reviewer #1: (No Response)

Reviewer #2: (No Response)

**Summary and General Comments**

Reviewer #1: (No Response)

Reviewer #2: (No Response)

PLOS authors have the option to publish the peer review history of their article (what does this mean?). If published, this will include your full peer review and any attached files.

Reviewer #1: No

Reviewer #2: No
---

## [Decision Letter · Decision Letter 1]

12 Apr 2024

Dear Dr. Chong,

Thank you very much for submitting your manuscript "Projection of dengue fever transmissibility under climate change in South and Southeast Asian countries" for consideration at PLOS Neglected Tropical Diseases. As with all papers reviewed by the journal, your manuscript was reviewed by members of the editorial board and by several independent reviewers. The reviewers appreciated the attention to an important topic. Based on the reviews, we are likely to accept this manuscript for publication, providing that you modify the manuscript according to the review recommendations. 

Sincerely,

Michael R Holbrook, PhD

Section Editor

Michael Holbrook

Section Editor

Reviewer's Responses to Questions

**Key Review Criteria Required for Acceptance?**

**Methods**

-Are the objectives of the study clearly articulated with a clear testable hypothesis stated?

-Is the study design appropriate to address the stated objectives?

-Is the population clearly described and appropriate for the hypothesis being tested?

-Is the sample size sufficient to ensure adequate power to address the hypothesis being tested?

-Were correct statistical analysis used to support conclusions?

-Are there concerns about ethical or regulatory requirements being met?

Reviewer #2: (No Response)

Reviewer #3: (No Response)

**Results**

-Does the analysis presented match the analysis plan?

-Are the results clearly and completely presented?

-Are the figures (Tables, Images) of sufficient quality for clarity?

Reviewer #2: (No Response)

Reviewer #3: (No Response)

**Conclusions**

-Are the conclusions supported by the data presented?

-Are the limitations of analysis clearly described?

-Do the authors discuss how these data can be helpful to advance our understanding of the topic under study?

-Is public health relevance addressed?

Reviewer #2: (No Response)

Reviewer #3: (No Response)

**Editorial and Data Presentation Modifications?**

Reviewer #2: (No Response)

Reviewer #3: The study collected data on weekly dengue fever (DF) incidence, mean daily temperature, and rainfall data at 30 locations in four countries - Malaysia, Singapore, Thailand, and Sri Lanka from 2012 to 2020, and used generalized additive and three Shared Socioeconomic Pathways to estimate the future peak and duration of DF transmission, which is critical for DF prevention and public health wellness.

Nevertheless, the manuscript requires minor revision due to the concerns raised below.

1 The authors chose 30 places from four nations for the relevant study; can these points from each country reflect all elements of the entire country due to geographical differences?

2 Climatic circumstances comprise a variety of elements, and the authors are confined in their study of the effect of climate on DF incidence to rainfall and temperature.

3. DF is a major public health concern. We should not only manage greenhouse gas emissions, but also implement more effective strategies and measures to eliminate DF, which should be discussed by the authors.

**Summary and General Comments**

Reviewer #2: The revised version adds good additional detail in the methods and results, and the changes in the Figures help to understand the results better. The authors did a nice job of address my initial comments and I have no further requested changes.

Reviewer #3: (No Response)

PLOS authors have the option to publish the peer review history of their article (what does this mean?). If published, this will include your full peer review and any attached files.

Reviewer #2: No

Reviewer #3: No

Figure Files:

Data Requirements:

Reproducibility:

References

---

## [Editor Report · Decision Letter 2]

19 Apr 2024

Dear Dr. Chong,

We are pleased to inform you that your manuscript 'Projection of dengue fever transmissibility under climate change in South and Southeast Asian countries' has been provisionally accepted for publication in PLOS Neglected Tropical Diseases.

Best regards,

Michael R Holbrook, PhD

Section Editor

Michael Holbrook

Section Editor

---

## [Editor Report · Acceptance letter]

24 Apr 2024

Dear Dr. Chong,

We are delighted to inform you that your manuscript, "Projection of dengue fever transmissibility under climate change in South and Southeast Asian countries," has been formally accepted for publication in PLOS Neglected Tropical Diseases.

Best regards,

Shaden Kamhawi

co-Editor-in-Chief

Paul Brindley

co-Editor-in-Chief
